**Data Availability Statement:** The data used for this study are not proprietary and will be made available by the SAMRC to individuals who want to access

# COVID-19 hospitalization and mortality and hospitalization-related utilization and expenditure: Analysis of a South African private health insured population

**Geetesh Solanki**[1,2], **Thomas Wilkinson**[2,3], **Shailav Bansal**[4], **Joshila Shiba**[4], **Samuel Manda**[5,6], **Tanya Doherty**[1,7] *

1 Health Systems Research Unit, South African Medical Research Council, Cape Town, South Africa, 2 Health Economics Unit, University of Cape Town, Cape Town, South Africa, 3 World Bank Group, Washington, DC, United States of America, 4 Quantium Health South Africa, Johannesburg, South Africa, 5 Department of Statistics, University of Pretoria, Pretoria, South Africa, 6 Biostatistics Research Unit, South African Medical Research Council, Pretoria, South Africa, 7 Department of Paediatrics and Child Health, University of Cape Town, Cape Town, South Africa

* tanya.doherty@mrc.ac.za

## Abstract

### Background

Evidence on the risk factors for COVID-19 hospitalization, mortality, hospital stay and cost of treatment in the African context is limited. This study aims to quantify the impact of known risk factors on these outcomes in a large South African private health insured population.

### Methods and findings

This is a cross sectional analytic study based on the analysis of the records of members belonging to health insurances administered by Discovery Health (PTY) Ltd. Demographic data for 188,292 members who tested COVID-19 positive over the period 1 March 2020–28 February 2021 and the hospitalization data for these members up until 30 June 2021 were extracted. Logistic regression models were used for hospitalization and death outcomes, while length of hospital stay and (log) cost per patient were modelled by negative binominal and linear regression models. We accounted for potential differences in the population served and the quality of care within different geographic health regions by including the health district as a random effect. Overall hospitalization and mortality risk was 18.8% and 3.3% respectively. Those aged 65+ years, those with 3 or more comorbidities and males had the highest hospitalization and mortality risks and the longest and costliest hospital stays. Hospitalization and mortality risks were higher in wave 2 than in wave 1. Hospital and mortality risk varied across provinces, even after controlling for important predictors. Hospitalization and mortality risks were the highest for diabetes alone or in combination with hypertension, hypercholesterolemia and ischemic heart disease.

the data for scientific and/or academic research purposes and are willing to commit to handling the data in a manner which is consistent with confidentiality requirements. The criteria for access would broadly incorporate requests from individuals with credible academic/research credentials. This does not alter our adherence to PLOS ONE policies on sharing data and materials. Requests for access to the data should be directed to: Mr Ishen Seocharan, SAMRC Email: Ishen. Seocharan@mrc.ac.za.

**Funding:** This study was supported by the South African Medical Research Council in the form of salaries for TD, GS, and SM.

**Competing interests:** I have read the journal's policy and the authors of this manuscript have the following competing interests: JS and SB are employees of Quantium Health

**Abbreviations:** CA, Cancer; COPD, Chronic Obstructive Pulmonary Disease; DM, Diabetes Mellitus; HIV, HIV; HC, Hypercholesterolemia; HT, Hypertension; HTH, Hyperthyroidism; IHD, Ischemic Heart Disease.

## Conclusions

These findings can assist in developing better risk mitigation and management strategies. It can also allow for better resource allocation and prioritization planning as health systems struggle to meet the increased care demands resulting from the pandemic while having to deal with these in an ever-more resource constrained environment.

## Introduction

COVID-19 was declared a Public Health Emergency of International Concern on 30 January 2020, and a pandemic on 11 March 2020 by the World Health Organization (WHO) [1,2]. There is now a body of evidence on the risk factors for COVID-19 related hospitalization and mortality [3–11] from studies in North America, Europe and China. A systematic review and meta-analysis of 102 papers covering 121,437 infected patients found comorbidities such as hypertension, diabetes, cardiovascular diseases, and chronic kidney disease were associated with the severity of COVID-19 infection [8]. Particularly, the elderly and males with underlying diseases were more likely to have severe COVID-19. Of the 102 papers included in the review, 80 were from Asia, 15 from Europe, 11 from North America and 1 from South America but none from Africa. A second systematic review and data synthesis of COVID-19 length of hospital stay across 52 studies, found that patients with COVID-19 in China remained in hospital longer than elsewhere [9]. None of the systematic reviews published to date have included any studies from Africa. The relevance of these studies to the broader African and sub-Saharan context is uncertain given the underlying demographic and disease profile differences between the regions [12].

Evidence on the risk factors in the African or South African context is limited with a review of the literature revealing only three published studies [13–15] which reported on the risk factors for COVID-19 related mortality. Two of these studies involved only public sector patients where HIV and tuberculosis are important risk factors for death. South African studies that have included data from private hospitals have included public vs private sector as a predictor variable rather than exploring outcomes separately in the two sectors (for example by stratifying on health care sector) [16].

Little is known about COVID-19 risk factors in the private sector population or how they may impact on hospital related utilization and expenditure patterns. The for-profit private sector is an important provider of health services in the sub-Saharan African region. A recent WHO report estimates the percentage of health services sourced from private providers in the AFRO region at 40% [17]. The South African health system is highly fragmented with substantial disparities in access, facilities and spending between the government-funded public health system and the private health system. Around 18% of the total South African population is covered by private voluntary health insurance [18] which is the predominant funding mechanism for the private health system. Access to private health services depends mainly on the ability to pay and there are stark racial and socio-economic differences in utilization of public compared to private health services. As of 2018, only 10% of black Africans were members of private medical schemes compared to 73% of white South Africans [19].

It is expected that there are differences in severe disease risk and hospitalization between the public and private sector populations and to date only two studies in South Africa have included data from the private sector [14,20] which found a lower overall case-fatality risk compared to the public sector but did not explore underlying risk factors in the two

populations. Evidence on the risk factors for COVID-19 related hospitalization, health care utilization and expenditure patterns are also limited and further evidence is required to confirm and better understand the patterns in this regard.

Relying largely on a fee for service model for provider payment and established clinical coding system, the private voluntary health insurance model generates substantial data enabling analysis of utilization, risk and expenditure of beneficiaries which can provide valuable insights. In South Africa, this approach has been taken to investigate for example, use of antibiotics [21], take-up of influenza vaccines [22] and caesarean section rates [23]. Elsewhere, researchers in the Republic of Korea have used insurance administrative datasets to investigate comorbidities and factors determining medical expenses and length of stay for COVID-19 patients admitted to hospital [24].

The aim of this study was to assess and quantify the impact of known risk factors on COVID-19 hospitalization, hospital related utilization and expenditure, and mortality in a large South African private health insured population over a 12 month period. The results from this study will contribute to addressing the gap in the knowledge base on the actual observed risks and subsequent hospitalization using real world data (RWD). It will enable targeted patient management strategies and risk stratification, identification of opportunities for provider quality and efficiency improvements and will generate information to assess the cost and cost effectiveness of preventative and treatment interventions for patients with COVID-19.

## Methods

### Study design

This is a cross sectional analytic study based on the analysis of the demographic and claims records of members belonging to medical schemes administered by Discovery Health (PTY) Ltd (DH), one of the largest health insurance administrators in South Africa.

### Study population

The study population consisted of 3.5 million individuals from 1.7 million households sharing the same health insurance policy. These policy holders belonged to 19 different health insurance schemes administered by DH, representing around a third of South Africa's privately insured population. The average family size, average contributions and health care expenditure of the study population were compared to that of the broader health insured population and found to be comparable ensuring that the findings of this study are generalisable to the broader South African population with private health insurance (S1 Table) [25].

### Data sources

Secondary de-identified demographic and claims data of members belonging to medical schemes administered by DH for the period from 1 March 2020 to 30 June 2021 was extracted from the data warehouse of Quantium Health, an independent company that provides data analytics and strategic consulting services to DH.

For each insured individual, the data contains the following variables: unique study individual identifier, date of birth, sex and province. For each claim submitted to the administrator for reimbursement of services rendered or items dispensed to an insured individual the data contains the following variables: a unique study individual identifier; dates for the commencement and completion of the service; a code and description for each service rendered/item dispensed, an ICD-10 (10th revision of the International Statistical Classification of Diseases and

Related Health Problems) code for the diagnosis of the condition being treated; a Current Procedural Terminology (CPT) code for the procedure carried out; a National Pharmaceutical Product Index (NAPPI) code for any surgical, medical or consumable item dispensed; and the amount being claimed.

## Data extraction and classification

The approach used to extract and classify the data is schematically summarized in Fig 1. From all the data for the period, a 3-step approach was used to extract the data. For the first step, individuals who had tested positive for COVID-19 through either the PCR, PKR or real-time RT-PCR tests in the period from 1 March 2020 to 28 February 2021 (study period) were identified and a "demographic extract" consisting of demographic, comorbidity and status elements was extracted for these individuals. For the comorbidity variable, the following conditions were considered as comorbidity risk factors for COVID-19 based on a review of published literature, consultation with the South African-based medical experts overseeing utilisation management at the health insurance, as well as consideration of the health profile of private sector patients in South Africa: Cancer, Chronic Renal Disease, Congestive Cardiac Failure, Chronic Obstructive Pulmonary Disease, Diabetes Mellitus, HIV, Hypercholesterolaemia, Hypertension, Hypothyroidism, Ischaemic Heart Disease, Pregnancy, Tuberculosis. The individuals with these comorbidities were identified using the South African Council for Medical Scheme Guideline algorithms for identifying members with medical conditions using claims records [26].

For the second step, the claims of the COVID-19 positive individuals over the period from 1 February 2020 to 30 June 2021 were assessed to determine whether they had been hospitalized for the treatment of COVID-19 and a hospital admission indicator was created and added to the demographic extract. Although only individuals who tested positive over the period from 1 March 2020 to 28 February 2021 were included in the study sample, the claims up to 30 June 2021 were included in hospitalization analysis to ensure that the data is not "right censored" as there is lag between testing and hospitalization and between hospitalization and the claims being received by the administrator.

For the third step, for all those COVID-19 positive individuals who had been hospitalized, a "hospital admissions" extract was created consisting of claims, length of stay and treatment marker elements.

To calculate the total cost per hospitalized COVID-19 patient we considered all claims for the hospitalization event including costs for pharmaceuticals, hospital bed charges, consumables, radiology services, general medical practitioner and specialist medical practitioner claims.

## Statistical analysis

Four COVID-19 outcomes were analyzed. Two of the outcomes were binary, namely a) whether the patient was hospitalized and b) whether the patient died (here, we assumed all deaths amongst these patients were due to or exacerbated by COVID-19). For the COVID-19 patients who were admitted to a hospital, two further outcomes were analyzed, namely c) length of stay (in days) in the hospital and (d) total cost of claims per patient.

The predictor variables considered in the analysis included age (at time of diagnosis); sex; number of commodities; pandemic wave: (pre-wave 1 (1 March 2020–6 June 2020), wave 1 (7 June 2020–22 August 2020), post wave 1 (23 August 2020–14 November 2020), Wave 2 (15 November 2020 –end Feb 2021)); province: (Eastern Cape, Free State, Gauteng, KwaZulu-Natal, Limpopo, Mpumalanga, North West, Northern Cape, Western Cape); health insurance

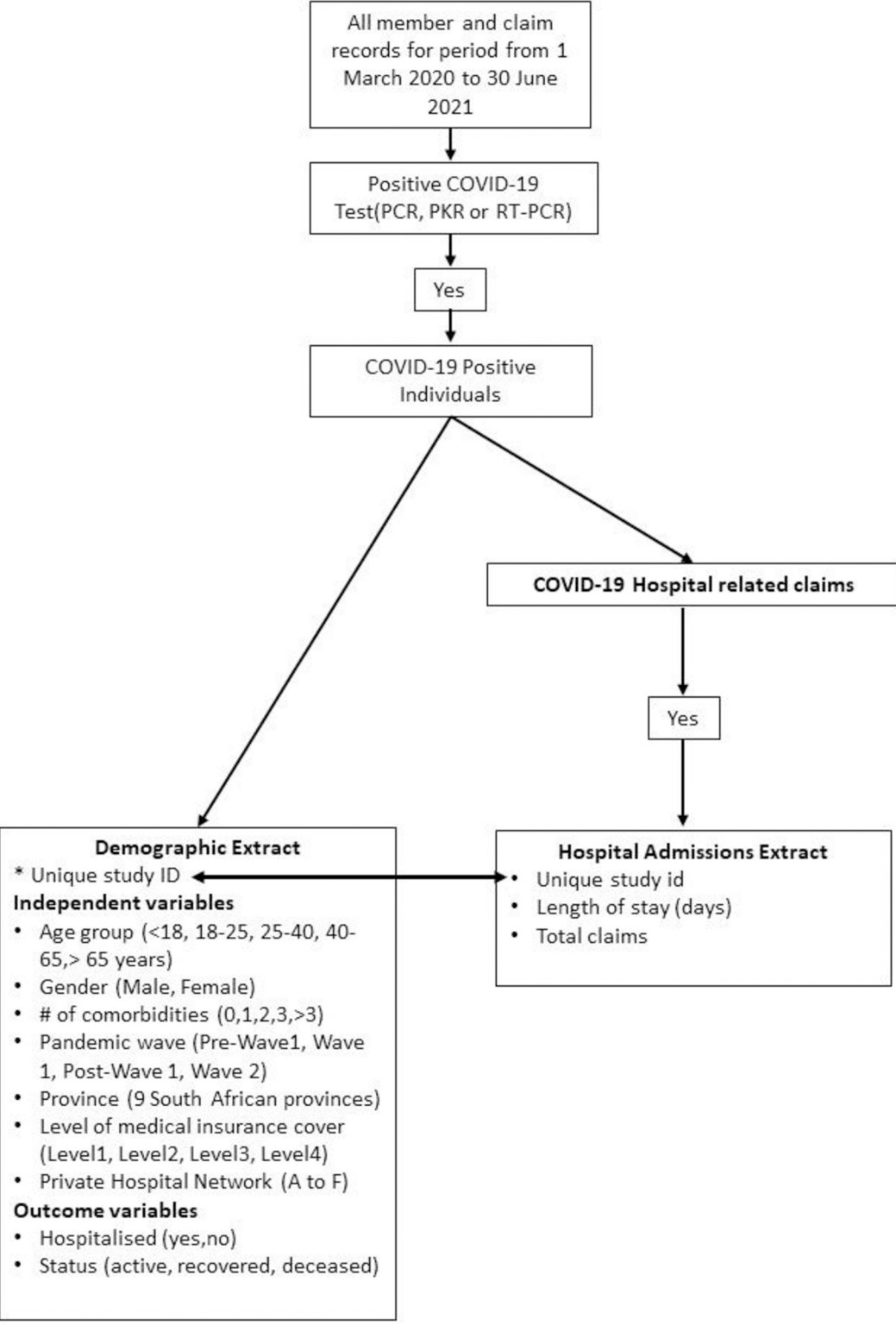

**Fig 1. Data extraction and classification.**

cover level: (1, 2, 3, 4 where cover level 1 plans offered the lowest level of benefits and cover level 4 plans offered the highest level of benefits) and hospital network: (the six main private hospital networks: A-F). A classification system was used for plans and provider networks to enable blinding of the actual names of plan types and specific hospital providers which is proprietary information. The data are grouped into 19 health regions for the insurance company administrative purposes. In our analyses, the health region was chosen for the random effects to account for potential differences in the population served and the quality of care within different geographic health regions.

Summary statistics included frequencies and percentages for categorical data, and for continuous data median and interquartile range were used. For modelling purposes, two-level random-effects logistic regression models were used for the two binary outcomes, where the level-2 unit was the health region.

Exploratory analysis showed that a Poisson model was insufficient to model the length of hospital stay as the data exhibited overdispersion in the sense that its variance exceeded its mean. Thus, a random effects negative binomial regression model was used for the number of days spent in a hospital and we accounted for health region variation as well as overdispersion. Unadjusted Incidence Rate Ratios (IRR) and adjusted Incidence Rate Ratios (aIRR) are presented.

The total cost data was heavily skewed to the right, and upon taking the logarithm of it, the transformed total cost had a "normal' shape. Thus, a linear mixed regression model on the log of total cost, again using the health region as a clustering level, was used.

Rather than presenting the estimated coefficients (which are increases in log costs per unit change in the respective predictor variable (category)), the estimated coefficients (e.g. beta1) were expressed as percentage increase or decrease depending on whether the coefficient is positive or negative using the formula, (exp(beta1)-1)x100. The coefficients from the linear regression model are shown in S2 Table.

We performed both univariate and multivariable analyses for all four outcomes (including all the predictors and random effects) to identify independent predictors of the modelled outcomes. The multivariate analyses produced adjusted effects as opposed to unadjusted effects from using univariate analysis.

A further separate analysis was carried out to assess the association between the most common comorbidities and combinations of comorbidities and the of risk of hospitalization and mortality. For this analysis, frequencies and percentages and unadjusted odds ratios (estimated using standard regression models) are reported for two outcome variables–hospitalization and mortality. STATA/SE 16.1 was used for all the analyses.

### Ethical considerations

Data for the study was made available as part of Quantium Health's commitment to support research initiatives with broader public health significance. The company does not advise its clients on the clinical treatment of its members. The data was accessed in terms of and under the conditions set out in the agreement between Quantium Health and Discovery Health and a memorandum of understanding between Quantium Health and the study investigators.

All the data was provided in a de-identified format and aggregated at an individual level and the research team had no access to information that would enable the identification of any individual. All findings are presented at an aggregate level and no confidential member, health care provider or scheme information is disclosed. Ethics approval for the use of the database to carry out this study was granted by the Ethics Committee of the SAMRC (project registration number EC018-4/2021).

## Results

### Sample description

The total dataset comprised the claims of an average total of 3,48 million individuals over the period 1 March 2020 to 28 February 2021. From this dataset, the claims data related to a total of 188,292 individuals who tested positive for COVID-19 over the period were extracted and analyzed. Of the total cases, 41.4% were aged between 40 and 65 years, 54.3% were female, 37.6% were diagnosed in Wave 1 and 51.1% were diagnosed in Wave 2, 40.9% were from Gauteng province, 65.6% had no comorbidities and 61.3% were on the Cover Level 3 plans (Table 1).

### Hospitalization risk

The overall hospitalization risk for COVID-19 positive individuals was 18.8% (Table 1). Age, sex and number of comorbidities were found to be independent predictors of hospital admission. Patients aged above 65 years (aOR 4.31; 95%CI 4.02–4.62); who were males (aOR 1.19; 95%CI 1.17–1.23) and had more than three comorbidities (aOR 3.97; 95% CI 3.76–4.21) were more likely to be admitted to hospital. Pre-wave 1 period (aOR 1.49; 95%CI 1.38–1.61), post-wave 1 (aOR 1.47; 95%CI 1.41–1.54), and wave 2 (aOR 1.18; 95%CI 1.15–1.21) all had a higher hospitalization risk compared to wave 1. Provincial differences in hospitalization risk were also observed with admissions more likely in Limpopo and the Northern Cape and less likely in Free State and Gauteng, compared with the Western Cape province. Health insurance cover level 4 (the most expensive plan with the highest level of insurance cover) was associated with a higher risk of hospitalization compared to plan level 1 (OR 1.22; 95% CI 1.16–1.27) in univariate analyses (Table 1). However, we did not include health insurance in the multivariable analyses because it was highly correlated with both age and number of comorbidities, which could have resulted in multicollinearity problems. Sixty-seven percent of individuals on plan level 4 were over the age of 40 and of those with more than 3 comorbidities almost a quarter (23%) were on plan level 4, compared with only 14% of those with 1 comorbidity.

### Mortality risk

The overall mortality risk for COVID-19 positive individuals was 3.3% (Table 1). In multivariable analysis, after adjustment for all other factors, mortality risk was the greatest for those aged above 65 year (aOR 108.26; 95% CI 63.85–183.57); males (aOR 1.61; 95% CI 1.52–1.69) and those with more than three comorbidities (aOR 2.64; 95% CI 2.41–2.89). Mortality risk was higher in pre-wave 1 (aOR 1.39; 95% CI 1.18–1.64) or Wave 2 (aOR 1.21; 95% CI 1.14–1.28) compared to wave 1. In terms of provincial differences, all provinces except Free State and Mpumalanga had a significantly higher risk of mortality compared to the Western Cape with the highest being KwaZulu-Natal Province (aOR 1.75; 95% CI 1.58–1.93) (Table 1). In univariate analysis medical insurance cover level 4 was associated with a higher risk of mortality compared to plan level 1 (OR 1.39; 95% CI 1.28–1.53) (Table 1).

### Hospitalization utilization

The overall median length of hospital stay for COVID-19 positive individuals was 6 days (IQR 3–10) (Table 2). In multivariable analysis, there was an increasing trend in length of hospital stay with age. Those aged over 65 years had a two-fold increased length of hospital stay compared with those less than 18 years (aIRR 2.00; 95% CI 1.89–2.12). Males had longer hospital stays than females (aIRR 1.08; 95% CI 1.06–1.09) and an increase in length of stay was observed for each additional comorbidity with individuals experiencing more than three

**Table 1. Sample description and univariate and multivariate analysis of factors associated with admission to hospital and mortality.**

| Variable | COVID-19 cases n (%) | Hospitalisation | | | Mortality | | |
|---|---|---|---|---|---|---|---|
| | | Hospitalised n (%) | Unadjusted OR (95% CI) | Adjusted OR (95% CI) | Deceased n (%) | Unadjusted OR (95% CI) | Adjusted OR (95% CI) |
| Age | | | | | | | |
| Less than 18 | 11,669 (6.2) | 1,204 (10.3) | Reference | Reference | 14 (0.1) | Reference | Reference |
| Between 18–25 | 11,932 (6.3) | 621 (5.2) | 0.48 (0.43–0.53) | 0.48 (0.44–0.53) | 13 (0.1) | 0.91 (0.43–1.93) | 0.95 (0.45–2.03) |
| Between 25–40 | 68,196 (36.2) | 7,357 (10.8) | 1.05 (0.98–1.12) | 1.03 (0.96–1.09) | 348 (0.5) | 4.27 (2.50–7.29) | 4.35 (2.55–7.43) |
| Between 40–65 | 78,031 (41.4) | 17,384 (22.3) | 2.49 (2.34–2.65) | 1.84 (1.73–1.96) | 2,674 (3.4) | 29.54 (17.46–49.96) | 23.12 (13.65–39.14) |
| Greater than 65 | 18,464 (9.8) | 8,901 (48.2) | 8.09 (7.57–8.64) | 4.31 (4.02–4.62) | 3,202 (17.3) | 174.66 (103.27–295.41) | 108.26 (63.85–183.57) |
| Sex | | | | | | | |
| Female | 102,184 (54.3) | 17,431 (17.1) | Reference | Reference | 2525 (2.5) | Reference | Reference |
| Male | 86,108 (45.7) | 18,036 (20.9) | 1.29 (1.26–1.32) | 1.19 (1.17–1.23) | 3726 (4.3) | 1.78 (1.69–1.88) | 1.61 (1.52–1.69) |
| Province | | | | | | | |
| Western Cape | 37,283 (19.9) | 7,248 (19.4) | Reference | Reference | 1,183 (3.2) | Reference | Reference |
| Eastern Cape | 11,962 (6.4) | 1,902 (15.9) | 0.78 (0.74–0.83) | 1.06 (0.93–1.20) | 461 (3.9) | 1.22 (1.09–1.36) | 1.48 (1.28–1.72) |
| Free State | 5,730 (3.0) | 1,078 (18.8) | 0.96 (0.89–1.03) | 0.86 (0.77–0.97) | 145 (2.5) | 0.79 (0.66–0.94) | 0.96 (0.77–1.19) |
| Gauteng | 76,249 (40.9) | 13,514 (17.7) | 0.89 (0.86–0.92) | 0.89 (0.82–0.96) | 2,278 (3.0) | 0.94 (0.87–1.01) | 1.26 (1.13–1.39) |
| KwaZulu-Natal | 34,856 (18.7) | 7,705 (22.1) | 1.17 (1.13–1.22) | 1.06 (0.97–1.17) | 1,604 (4.6) | 1.47 (1.36–1.59) | 1.75 (1.58–1.93) |
| Limpopo | 4,091 (2.2) | 717 (17.5) | 0.88 (0.81–0.96) | 1.21 (1.07–1.38) | 114 (2.8) | 0.87 (0.72–1.06) | 1.48 (1.19–1.84) |
| Mpumalanga | 7,096 (3.8) | 1,279 (18.0) | 0.91 (0.85–0.97) | 0.99 (0.90–1.11) | 184 (2.6) | 0.81 (0.69–0.95) | 1.13 (0.95–1.35) |
| North West | 6,184 (3.3) | 1,316 (21.3) | 1.12 (1.05–1.12) | 1.04 (0.93–1.17) | 186 (3.0) | 0.95 (0.81–1.11) | 1.29 (1.08–1.54) |
| Northern Cape | 3,038 (1.6) | 599 (19.7) | 1.02 (0.93–1.12) | 1.14 (1.02–1.29) | 79 (2.6) | 0.81 (0.65–1.03) | 1.36 (1.06–1.74) |
| # of comorbidities | | | | | | | |
| 0 | 123,507 (65.6) | 15,110 (12.2) | Reference | Reference | 1,757 (1.4) | Reference | Reference |
| 1 | 30,668 (16.3) | 6,758 (22.0) | 2.03 (1.96–2.09) | 1.52 (1.47–1.57) | 1,100 (3.6) | 2.58 (2.39–2.78) | 1.29 (1.19–1.40) |
| 2 | 17,105 (9.1) | 5,744 (33.6) | 3.63 (3.49–3.76) | 2.19 (2.11–2.28) | 1,252 (7.3) | 5.47 (5.08–5.89) | 1.79 (1.65–1.94) |
| 3 | 10,382 (5.5) | 4,243 (40.9) | 4.96 (4.75–5.17) | 2.64 (2.52–2.77) | 1,098 (10.6) | 8.19 (7.58–8.86) | 2.13 (1.96–2.33) |
| >3 | 6,630 (3.5) | 3,612 (54.5) | 8.58 (8.16–9.04) | 3.97 (3.76–4.21) | 1,044 (15.7) | 12.96 (11.94–14.04) | 2.64 (2.41–2.89) |
| Pandemic wave | | | | | | | |
| Pre-Wave 1 | 4,882 (2.59) | 1,091 (22.3) | 1.42 (1.33–1.53) | 1.49 (1.38–1.61) | 186 (3.8) | 1.35 (1.15–1.57) | 1.39 (1.18–1.64) |
| Wave 1 | 70,742 (37.57) | 11,884 (16.8) | Reference | Reference | 2,021 (2.9) | Reference | Reference |
| Post-wave 1 | 16,475 (8.75) | 3,662 (22.2) | 1.41 (1.36–1.47) | 1.47 (1.41–1.54) | 386 (2.3) | 0.81 (0.73–0.91) | 0.72 (0.64–0.81) |
| Wave 2 | 96,193 (51.09) | 18,380 (19.6) | 1.20 (1.17–1.24) | 1.18 (1.15–1.21) | 3,658 (3.8) | 1.34 (1.27–1.42) | 1.21 (1.14–1.28) |
| Medical insurance cover level | | | | | | | |
| Level 1 | 26,512 (14.3) | 4,796 (18.1) | Reference | | 856 (3.2) | Reference | |
| Level 2 | 22,460 (12.1) | 4,554 (20.3) | 1.15 (1.10–1.20) | | 804 (3.6) | 1.11 (1.01–1.23) | |
| Level 3 | 113,721 (61.3) | 21,088 (18.5) | 1.03 (0.99–1.07) | | 3,531 (3.1) | 0.96 (0.89–1.04) | |
| Level 4 | 22,835 (12.3) | 4833 (21.2) | 1.22 (1.16–1.27) | | 1,019 (4.5) | 1.39 (1.28–1.53) | |
| Total | 188,292 | 35,467 (18.84) | | | 6,251 (3.32) | | |

OR = odds ratio.

comorbidities having the longest period of hospitalization (aIRR 1.14; 95% CI 1.11–1.18). Length of hospital stay was longer in wave 2 compared to wave 1 (aIRR 1.03; 95% CI 1.01–1.05). Provincial differences in length of hospitalization were observed with Eastern Cape, Free State, Gauteng, KwaZulu-Natal and North West all having significantly longer median length of hospitalization of COVID-19 patients compared to the Western Cape (Table 2).

**Table 2. Univariate and multivariate analysis of factors associated with hospitalisation length of stay.**

| Hospital utilisation (days) | | | | |
|---|---|---|---|---|
| Variable | Total hospitalised COVID-19 cases | Median length of hospitalisation (IQR) | Unadjusted length of hospitalisation IRR (95% CI) | Adjusted length of hospitalisation aIRR (95% CI) |
| Age | | | | |
| Less than 18 | 1,204 | 3 (1–4) | Reference | Reference |
| Between 18–25 | 621 | 3 (2–5) | 1.13 (1.03–1.23) | 1.09 (1.00–1.19) |
| Between 25–40 | 7,357 | 4 (2–7) | 1.34 (1.26–1.41) | 1.36 (1.28–1.44) |
| Between 40–65 | 17,384 | 7 (4–11) | 2.12 (2.01–2.24) | 1.80 (1.71–1.90) |
| Greater than 65 | 8,901 | 8 (4–14) | 2.54 (2.41–2.69) | 2.00 (1.89–2.12) |
| Sex | | | | |
| Female | 17,431 | 5 (3–10) | Reference | Reference |
| Male | 18,036 | 6 (4–11) | 1.17 (1.14–1.19) | 1.08 (1.06–1.09) |
| Province | | | | |
| Western Cape | 7,248 | 5 (3–10) | Reference | Reference |
| Eastern Cape | 1,902 | 6 (3–11) | 1.05 (1.01–1.10) | 1.09 (1.03–1.15) |
| Free State | 1,078 | 6 (3–10) | 1.03 (0.97–1.01) | 1.08 (1.01–1.15) |
| Gauteng | 13,514 | 6 (3–11) | 1.05 (1.03–1.08) | 1.09 (1.06–1.14) |
| KwaZulu-Natal | 7,705 | 7 (4–11) | 1.08 (1.05–1.11) | 1.14 (1.09–1.18) |
| Limpopo | 717 | 5 (3–8) | 0.87 (0.81–0.93) | 0.97 (0.90–1.05) |
| Mpumalanga | 1,279 | 5 (3–10) | 0.99 (0.94–1.04) | 1.02 (0.97–1.08) |
| North West | 1,316 | 7 (3–11) | 1.06 (1.00–1.11) | 1.11 (1.05–1.18) |
| Northern Cape | 599 | 5 (3–8) | 0.89 (0.82–0.96) | 0.99 (0.92–1.06) |
| # of comorbidities | | | | |
| 0 | 15,110 | 5 (3–9) | Reference | Reference |
| 1 | 6,758 | 6 (3–11) | 1.25 (1.22–1.28) | 1.08 (1.06–1.11) |
| 2 | 5,744 | 7 (4–12) | 1.39 (1.36–1.43) | 1.13 (1.10–1.16) |
| 3 | 4,243 | 7 (4–13) | 1.44 (1.39–1.48) | 1.11 (1.08–1.14) |
| >3 | 3,612 | 8 (4–13) | 1.55 (1.50–1.60) | 1.14 (1.11–1.18) |
| Pandemic wave | | | | |
| Pre-Wave 1 | 1,091 | 6 (3–12) | 1.09 (1.03–1.14) | 1.06 (1.01–1.11) |
| Wave 1 | 11,884 | 6 (3–11) | Reference | Reference |
| Post-wave 1 | 3,662 | 5 (2–9) | 0.88 (0.85–0.90) | 0.89 (0.87–0.93) |
| Wave 2 | 18,380 | 6 (3–11) | 1.31 (1.27–1.35) | 1.03 (1.01–1.05) |
| Medical insurance cover level | | | | |
| Level 1 | 4,796 | 6 (3–10) | Reference | |
| Level 2 | 4,554 | 6 (3–10) | 1.00 (0.96–1.04) | |
| Level 3 | 21,088 | 6 (3–10) | 1.00 (0.97–1.03) | |
| Level 4 | 4,833 | 7 (3–12) | 1.11 (1.07–1.15) | |
| Private Hospital network | | | | |
| Network A | 1,307 | 6 (4–10) | Reference | |
| Network B | 9,601 | 7 (4–11) | 1.09 (1.03–1.15) | |
| Network C | 546 | 5 (3–9) | 0.77 (0.70–0.84) | |
| Network D | 5,407 | 6 (3–11) | 1.04 (0.98–1.09) | |
| Network E | 9,151 | 5 (3–10) | 0.93 (0.88–0.98) | |
| Network F | 8,846 | 6 (3–11) | 1.02 (0.97–1.07) | |
| Total | 35,467 | 6 (3–10) | | |

aIRR = adjusted incidence risk ratio.

In the unadjusted model, assessing the effect of insurance cover, only insurance plan level 4 (the most expensive plan with the highest level of cover), was significantly associated with length of hospital stay but this effect was not significant in the adjusted model due to the variable being highly correlated with age and number of comorbidities. In univariate analysis, hospital network B had significantly longer hospital stays compared with network A hospitals, but the effect was not significant in the multivariable model.

## Hospitalization expenditure

The overall median hospitalization cost per COVID-19 positive case was R49,836 (IQR R28,464—R107,020) (Table 3). After adjustment for all other factors, there was an increasing hospitalization cost with each age category and those over age 65 years incurred a 172% increased cost of hospitalization compared to individuals under age 18 years (95% CI 153.45% - 191.54%). The cost of hospitalization for males was 18% higher than that for females (95% CI 16.18%– 20.92%). Cost of hospitalization increased for each additional comorbidity. Those with more than three comorbidities had 28% higher hospitalization costs compared with individuals with no comorbidities (95% CI 23.37%– 33.64%). With regard to pandemic wave period, hospitalization during wave 2 was 7% more costly compared to the wave 1 period (95% CI 4.08% - 9.42%). With regard to provincial differences, Gauteng and KwaZulu-Natal were both significantly more costly than the Western Cape (11% and 4% more costly respectively), whilst Limpopo and the Northern Cape were less costly compared to the Western Cape (14% and 10% less costly respectively). Hospitalization cost differences were noted between insurance plan levels, with plans 2, 3 and 4 being 8% more costly compared with level 1 plans We also observed differences in cost across hospital networks after adjusting for all other covariates. Hospital networks C, D, E and F were all significantly less costly compared to network A (27%, 12%, 17% and 12% less costly respectively) (Table 3).

## Risk by comorbidity condition type

Of the conditions considered as comorbidity factors, Diabetes Mellitus (on its own or in combination with other comorbidities) carried the highest hospitalization risk (OR 3.6; 95% CI 3.27–3.94 for Diabetes Mellitus only; OR 6.6; 95%CI 5.88–7.43 for Diabetes with hypertension, hypercholesterolemia and Ischemic heart disease) (Table 4). In terms of mortality risk, the combination of diabetes with hypertension, hypercholesterolemia and Ischemic heart disease carried the highest mortality risk (OR 10.25; 95% CI 8.57–12.27). Hypertension in combination with heart disease (OR 6.94; 95% CI 5.66–8.51) or cancer (OR 6.10; 95% CI 4.47–8.33) also carried an increased risk for mortality (Table 4).

## Discussion

This is the first study describing risk factors for COVID-19 hospitalization and mortality and hospitalization related utilization and expenditure amongst a private health insured population in Africa. From a study population of 188,292 COVID-19 cases, we found overall hospitalization rates and mortality rates of 18.8% and 3.3% respectively.

COVID-19 positive individuals above the age of 65 years, those with 3 or more comorbidities and males had the highest risk across all 4 outcome measures. Overall, in line with studies carried out elsewhere [24], the findings suggest that the strongest predictors for COVID-19 related hospitalization, mortality [10,11], hospital related utilization [27] and expenditure [28] was age, followed by the number of comorbidities and then sex.

Regarding specific comorbidities, diabetes alone or in combination with hypertension, hypercholesterolemia and ischemic heart disease carried the greatest risk for hospitalization

**Table 3. Univariate and multivariate analysis of factors associated with hospitalisation cost.**

Cost per hospitalised COVID-19 patient (SA Rands)

| Variable | Total hospitalised COVID-19 cases | Median cost/ patient (IQR) | Unadjusted % Difference (Increase/ decrease in cost) (95%CI) | Adjusted % Difference (Increase/ decrease in cost) (95%CI) |
|---|---|---|---|---|
| Age | | | | |
| Less than 18 | 1,204 | 24,400 (15,552–39,586) | Reference | Reference |
| Between 18–25 | 621 | 30,390 (18,944–54,741) | 20.92 (9.42, 33.64) | 27.12 (13.88, 40.49) |
| Between 25–40 | 7,357 | 42,051 (24,974–68,435) | 61.61 (52.20, 73.33) | 60.00 (50.68, 71.60) |
| Between 40–65 | 17,384 | 57,801 (33,276–138,005) | 158.57 (143.51, 174.56) | 131.64 (118.15, 145.96) |
| Greater than 65 | 8,901 | 83,335 (41,290–203,098) | 215.82 (9.42, 235.35) | 171.83 (153.45, 191.54) |
| Sex | | | | |
| Female | 17,431 | 50,162 (28,650–103,434) | Reference | Reference |
| Male | 18,036 | 60,793 (33,151–155,153) | 27.12 (23.37, 29.69) | 18.53 (16.18, 20.92) |
| Province | | | | |
| Western Cape | 7,248 | 47,080 (26,290–103,145) | Reference | Reference |
| Eastern Cape | 1,902 | 48,843 (28,421–101,642) | 1.01 (-3.92, 7.25) | -1.00 (-5.82, 5.13) |
| Free State | 1,078 | 45,547 (27,073–84,580) | -5.82 (-12.19, 1.01) | -1.98 (-7.69, 5.13) |
| Gauteng | 13,514 | 54,059 (30,663–117,780) | 15.03 (11.63, 18.53) | 11.63 (8.33, 15.03) |
| KwaZulu-Natal | 7,705 | 50,388 (29,753–107,514) | 9.42 (5.13, 12.75) | 4.08 (1.01, 8.33) |
| Limpopo | 717 | 36,514 (21,760–73,986) | -20.55 (-26.66, -13.06) | -14.79 (-21.34, -7.69) |
| Mpumalanga | 1,279 | 41,264 (23,406–91,366) | -10.42 (-16.47, -4.88) | -3.92 (-10.42, 2.02) |
| North West | 1,316 | 47,634 (27,784–88,583) | -1.00 (-7.69, 5.13) | -0.10 (-5.82, 6.18) |
| Northern Cape | 599 | 39,213 (23,116–69,723) | -19.75 (-26.65, -12.19) | -10.42 (-17.30, -1.98) |
| # of comorbidities | | | | |
| 0 | 15,110 | 42,079 (24,547–76,712) | Reference | Reference |
| 1 | 6,758 | 50,359 (29,152–107,048) | 24.61 (20.92, 28.40) | 7.25 (4.08, 11.63) |
| 2 | 5,744 | 58,489 (33,018–132,798) | 46.23 (40.49, 50.68) | 17.35 (13.88, 20.92) |
| 3 | 4,243 | 62,213 (34,300–148,067) | 55.27 (50.68, 61.61) | 19.72 (16.18, 24.61) |
| >3 | 3,612 | 73,423 (37,685–170,703) | 73.33 (66.53, 80.40) | 28.40 (23.37, 33.64) |
| Pandemic wave | | | | |
| Pre-Wave 1 | 1,091 | 52,011 (27,318–120,072) | 6.18 (-0.40, 13.88) | 13.88 (6.18, 20.92) |
| Wave 1 | 11,884 | 48,577 (27,537–103,472) | Reference | Reference |

(*Continued*)

**Table 3.** (Continued)

**Cost per hospitalised COVID-19 patient (SA Rands)**

| Variable | Total hospitalised COVID-19 cases | Median cost/ patient (IQR) | Unadjusted % Difference (Increase/decrease in cost) (95%CI) | Adjusted % Difference (Increase/decrease in cost) (95%CI) |
|---|---|---|---|---|
| Post-wave 1 | 3,662 | 45,169 (25,009–90,903) | -10.42 (-13.93, -6.76) | -2.96 (-6.76, 1.01) |
| Wave 2 | 18,380 | 51,524 (29,882–112,648) | 7.25 (5.13, 10.52) | 7.25 (4.08, 9.42) |
| Medical insurance cover level | | | | |
| Level 1 | 4,796 | 44,071 (24,695–91,445) | Reference | Reference |
| Level 2 | 4,554 | 50,698 (29,034–114,502) | 20.92 (15.03, 25.86) | 8.65 (4.08, 13.43) |
| Level 3 | 21,088 | 49,326 (28,477–102,979) | 16.18 (12.75, 19.72) | 8.76 (5.13, 12.75) |
| Level 4 | 4,833 | 59,609 (32,486–131,414) | 37.71 (32.31, 43.33) | 8.65 (4.08, 13.43) |
| Private Hospital network | | | | |
| Network A | 1,307 | 53,407 (30,403–123,034) | Reference | Reference |
| Network B | 9,601 | 59,086 (35,733–128,490) | 11.63 (5.13, 18.53) | 5.13 (-1.00, 11.63) |
| Network C | 546 | 36,068 (22,360–69,732) | -35.60 (-42.31, -28.11) | -27.39 (-34.30, -18.94) |
| Network D | 5,407 | 49,165 (27,202–101,265) | -12.19 (-17.30, -5.82) | -12.19 (-17.30, -6.76) |
| Network E | 9,151 | 44,706 (25,529–91,528) | -17.30 (-22.12, -12.19) | -17.30 (-22.12, -12.19) |
| Network F | 8,846 | 47,670 (27,554–105,322) | -10.42 (-15.63, -3.92) | -12.19 (-17.30, -6.76) |
| Total | 35,467 | 49,836 (28,464–107,020) | | |

and death. These comorbidity risk factors for severe disease and death are similar to other settings. In contrast to research amongst public sector COVID-19 patients in South Africa [13], we did not find an association between HIV and mortality reflecting the different underlying disease profile of the private sector population. Around 4.7% of the private insured population in South Africa are registered on an HIV management program [25] whilst the HIV prevalence rate in the general population is 14% [29]. Another recent study in South Africa exploring risk factors for COVID-19 related in-hospital mortality found an HIV prevalence amongst hospitalized Covid-19 patients of 20.4% in the public sector and 2.2% in the private sector [16].

Research based on data from the national surveillance system, including both public and private sector patients, has reported a case fatality risk of 18.7% amongst hospitalized COVID-19 patients in the private sector and 27.5% amongst public sector patients [14]. Our mortality risk does include some deaths (509) amongst individuals who were never hospitalized although almost all (92%) of the deaths in our sample occurred in hospital. Amongst hospitalized cases in our study the mortality risk is 16% (5742/35467) which compares well to the rate reported from national surveillance amongst private sector COVID-19 admissions. Differences in mortality risk between the public and private sector are expected due to differences in underlying disease profiles of patients, resourcing and case load differences.

**Table 4. Hospital risk and mortality risk by co-morbidity condition combinations.**

| Condition/Condition combinations | Total cases | Hospitalised n (%) | Unadjusted OR (95% CI) | Deaths n(%) | Unadjusted OR |
|---|---|---|---|---|---|
| No Co-morbidities | 123,507 | 15,110 (12.2) | 1.0(0.98–1.02) | 1,757 (1.4) | 1.00(0.94–1.07) |
| Other co-morbid combinations | 16,018 | 7,034 (43.9) | 5.6 (5.42–5.82) | 1,844 (11.5) | 9.01 (8.42–9.64) |
| DM-HT-HC-IHD | 1,141 | 547 (47.9) | 6.6 (5.88–7.43) | 147 (12.9) | 10.25 (8.57–12.27) |
| DM-HT-HC | 4,578 | 1,876 (41.0) | 5.0 (4.68–5.29) | 449 (9.8%) | 7.54 (6.77–8.4) |
| DM-HT | 2,601 | 1,061(40.8) | 4.9 (4.56–5.35) | 287 (11.0) | 8.59 (7.53–9.8) |
| HT-HC-IHD | 1,176 | 428 (36.4) | 4.1 (3.64–4.62) | 107 (9.1) | 6.94 (5.66–8.51) |
| DM-HC | 1,484 | 532 (35.8) | 4.0 (3.6–4.46) | 85 (5.7) | 4.21 (3.37–5.27) |
| HT-CA | 544 | 187 (34.4) | 3.8 (3.15–4.49) | 44 (8.1) | 6.10 (4.47–8.33) |
| DM | 2,085 | 695 (33.3) | 3.6 (3.27–3.94) | 102 (4.9) | 3.56 (2.9–4.37) |
| CA | 1,034 | 325 (31.4) | 3.3 (2.88–3.76) | 61 (5.9) | 4.34 (3.34–5.65) |
| HIV-HT | 755 | 22 (9.7) | 3.0 (2.59–3.55) | 44 (5.8) | 4.29 (3.15–5.84) |
| HT-HC | 3,849 | 1,133 (29.4) | 3.0 (2.78–3.21) | 268 (7.0) | 5.19 (4.54–5.93) |
| HT-HTH | 1,067 | 297 (27.8) | 2.8 (2.42–3.17) | 67 (6.3) | 4.64 (3.61–5.97) |
| HT | 14,167 | 3,609 (25.5) | 2.5 (2.35–2.55) | 703 (5.0) | 3.62 (3.31–3.96) |
| HC | 2,201 | 442 (20.1) | 1.8 (1.62–2) | 56 (2.5) | 1.81 (1.38–2.37) |
| HIV | 4,541 | 806 (17.7) | 1.6 (1.43–1.68) | 93 (2.0) | 1.45 (1.17–1.79) |
| COPD | 6,009 | 941 (15.7) | 1.3 (1.24–1.43) | 107 (1.8) | 1.26 (1.03–1.53) |
| HTH | 1,537 | 220 (14.3) | 1.2 (1.04–1.39) | 30 (2.0) | 1.38 (0.96–1.99) |

Provincial variation in all four outcome measures were found, even after adjustment for all other factors. This reflects differences in clinical practice between private hospitals which may not necessarily adhere to national Department of Health COVID-19 protocols. There may also be underlying differences in health seeking behaviour across provinces and different thresholds applied by general practitioners regarding when to admit patients to hospital. With regard to differences in hospitalization cost, Gauteng was almost 12% more expensive compared to the Western Cape. This is likely due to the reduced plan costs for coastal versus inland hospitals under the DH plan options [30].

We found higher rates of hospitalization and mortality and longer duration of hospital stay during wave 2 compared with wave 1. This is similar to findings from the national surveillance system and is likely due to the higher incidence of COVID-19, greater pressure on hospitals and the emergence of the Beta variant [14].

In unadjusted analysis, individuals on level 4 insurance plans were found to have an increased risk of both hospitalization and mortality. This was related to individuals on these plans being older and with more comorbidities reflecting the trend for individuals to buy more expensive and comprehensive medical insurance as they get older and sicker [31].

Hospital networks also differed in the cost of COVID-19 care after adjustment for risk factors. While this could be due to differences in the reimbursement rates across the various hospital network groups, variation in the underlying clinical management of patients across hospital groups is likely to have played a role.

This analysis has a number of limitations. Only services for which claims were submitted were analysed. This could result in under-recording of the COVID-19 cases, particularly of "milder" cases and could result in an overstatement of the reported outcome measures. As this data set is from a period prior to the mass distribution of home-based COVID-19 self-test kits, COVID testing was by doctor referral available under private insurance at no charge where the patient tested positive. For those paying out of pocket the test was relatively expensive (US$55 per test). It is possible that some patients with private voluntary insurance elected to seek

testing in the public system or pay out of pocket for the test in which case they would be missed from the data set. However, we consider that this number would be very small as there was strong incentive for those insured to utilize their insurance benefits, and the requirement for a doctor referral also strengthened capture of data. We further note that (1) our study population was limited to those who tested COVID-19 positive and (2) all the statistics that we present are based on those who tested positive (not the entire insured population).

In addition, the costs do not reflect additional out of pocket expenditure for claims not covered under the benefits of certain health plans, for example the use of the pharmaceutical Remdesivir as an adjunct to existing treatments covered by the insurance plan.

Obesity and smoking, identified as risk factors in other studies, could not be included in our analysis due to limited or incomplete coding of these risks in our dataset. We have however included the co-morbidity chronic obstructive pulmonary disease and other large analyses that have included both smoking status and chronic pulmonary disease in adjusted models have noted a mediating effect which limits the ability to explore the independent association of smoking status [32]. Vaccination status could also not be included due to the timeframe for the analysis in relation to the vaccine roll-out. Vaccination roll-out in South Africa began in May 2021 with individuals over the age of 60 years and our analysis included hospitalization data up until 20 June 2021.

The potential for overcrowding in health facilities leading to increased death due to COVID-19 represented a significant concern for South Africa's COVID-19 response. Differentiating risk by pandemic wave period allowed this analysis to partially account for impact due to overcrowding, however it is unknown whether overcrowding had a differentiated impact on COVID-19 outcomes contingent upon particular risk factors.

This analysis has significant policy implications for both private and public sectors in South Africa for targeted, risk-based interventions and for reducing unwarranted variation. The outputs from the study can provide a basis for developing "risk calculators" to enable providers and funders to develop risk-based management strategies. Risk information can also inform broader policy, including risk stratification for employer "work from home" policies and other initiatives to reduce risk of infection.

The finding of provincial and hospital group variation on outcomes after adjusting for other risk factors is in line with the findings of the Competition Commission's Health Market Inquiry which identified this variation as a major source of private sector inefficiency in South Africa [33]. The results highlight the difficulties related to efforts to contain health system costs, with complex dynamics between independent clinician judgment, hospital groups, and insurance plan types in a system with few mechanisms for standardization, and even built-in efficiency impediments–for example health insurance providers are by law required to negotiate separately with individual hospital groups. They also highlight the need to identify and minimize unwarranted variation through the implementation of protocols which are evidence based, effective and cost-effective across the private sector. In addition, the analysis provides a basis for determining the cost effectiveness of different treatment and preventative interventions–an understanding of the expected cost and mortality risk in the South African context for different patient populations following a COVID-19 diagnosis enables realistic estimations about how best limited resources can be used in developing clinical guidelines and protocols.

Finally, this analysis demonstrates the untapped potential of RWD in health policy decision making and planning. Through the use of relatively simple multiple regression analysis of a substantive data set, insights were achievable in terms of variation on a range of outcomes, much of which would be unachievable in even a large-scale clinical trial. Using more extensive data analysis and time series, this dataset may also enable testing of various interventions and disease management dynamics, most notably the impact of vaccination and other treatment

strategies. This analysis was conducted with ethics approval and maintained confidentiality of individual patient data which demonstrates a workable model of analysis. The larger lesson for health systems is that data systems across public and private sectors must be improved to enable use of data and analytics in decision making. Traditional analytical approaches for informing health investment and planning decisions rely on modeled prevalence and epidemiological information with intervention effects from published literature. While this approach enables evidence-informed decision making, it will always be limited to the extent that it reflects the actual context of the health system, inefficiencies and variations. The advancing digitization of health systems means that the routine use of health system information for investment, planning, efficiency and quality improvement is possible if the appropriate structures are present.

## Conclusion

The information from this study, with one of the largest private sector patient datasets, can assist in developing better risk mitigation and management strategies. It can also allow for better resource allocation planning and prioritization strategies as health care systems struggle to meet the immediate and longer term increased health care demands resulting from the COVID-19 pandemic while having to deal with these in an ever-more resource constrained environment [34].

## Supporting information

**S1 Table. Comparison of Discovery Health administration profile versus rest of the rest of the insured population in South Africa.**
(DOCX)

**S2 Table. Univariate and multivariate analysis of factors associated with hospitalisation cost with coefficients.**
(DOCX)

## Acknowledgments

We acknowledge the assistance of staff at Quantium Health for the extraction of data.

## Author Contributions

**Conceptualization:** Geetesh Solanki, Shailav Bansal, Joshila Shiba, Samuel Manda, Tanya Doherty.

**Data curation:** Thomas Wilkinson, Shailav Bansal, Joshila Shiba.

**Formal analysis:** Geetesh Solanki, Thomas Wilkinson, Samuel Manda, Tanya Doherty.

**Writing – original draft:** Geetesh Solanki, Tanya Doherty.

**Writing – review & editing:** Geetesh Solanki, Thomas Wilkinson, Joshila Shiba, Samuel Manda.

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
