## [Decision Letter · Decision Letter 0]

22 Feb 2022

PONE-D-21-33875COVID-19 hospitalization and mortality and hospitalization-related utilization and expenditure: Analysis of a South African private health insured population.PLOS ONE

Dear Dr. Doherty,

Thank you for submitting your manuscript to PLOS ONE. After careful consideration, we feel that it has merit but does not fully meet PLOS ONE’s publication criteria as it currently stands. Therefore, we invite you to submit a revised version of the manuscript that addresses the points raised during the review process.

We look forward to receiving your revised manuscript.

Kind regards,

Raymond Nienchen Kuo, Ph.D

Academic Editor

PLOS ONE

Journal Requirements:

3. We note that you have stated that you will provide repository information for your data at acceptance. Should your manuscript be accepted for publication, we will hold it until you provide the relevant accession numbers or DOIs necessary to access your data. If you wish to make changes to your Data Availability statement, please describe these changes in your cover letter and we will update your Data Availability statement to reflect the information you provide

Reviewers' comments:

Reviewer's Responses to Questions

**Comments to the Author**

1. Is the manuscript technically sound, and do the data support the conclusions?

Reviewer #1: Partly

Reviewer #2: Partly

2. Has the statistical analysis been performed appropriately and rigorously? 

Reviewer #1: I Don't Know

Reviewer #2: I Don't Know

3. Have the authors made all data underlying the findings in their manuscript fully available?

Reviewer #1: No

Reviewer #2: No

4. Is the manuscript presented in an intelligible fashion and written in standard English?

Reviewer #1: Yes

Reviewer #2: Yes

5. Review Comments to the Author

Reviewer #1: The authors present a study looking at the risk factors for COVID-19 hospitalization, mortality, hospital stay and cost of treatment in South Africa.

I think this is an important study which adds a lot of value, particularly as it covers a region of the world for which there has been limited information. The introduction is well written and interesting. It clearly identifies the research question and highlights the importance of this study. It is a rich dataset, which covers a long period of time, not just the beginning of the pandemic, allowing for changes in hospitalisation, mortality, hospital length and cost to be observed. However, I do have several concerns about the methodology which I have outlined below. Unfortunately, Tables 1-4 also seem to be missing from the manuscript, without which it makes it very difficult to assess the results. I have done my best without them, but would appreciate if a copy of the tables could be sent to me so that I can review the results properly. I also think there lacks interpretation and contextualisation of the results in the discussion (highlighted below).

I think if the authors could address these concerns methodological concerns, and expand the interpretation in the discussion, it is a very worthwhile paper.

1. In your population, do you know all people who have received a positive test? Or is it possible that individuals could have received a positive test from a different source, and this is not recorded in the data? This is an important point and should be made clear in the paper. If it is possible that positive tests could have been missed, then this warrants further discussion (i.e. if those individuals did not need hospitalization, this would over-estimate the estimates of hospitalisation and mortality).

2. I think there are comorbidities that are known risk factors for COVID-19 which have not been included in the study.

You state in the methods:

“Conditions were considered as comorbidity risk factors for COVID-19 based on a review of published literature: Cancer, Chronic Renal Disease, Congestive Cardiac Failure, Chronic Obstructive Pulmonary Disease, Diabetes Mellitus, HIV, Hypercholesterolaemia, Hypertension, Hypothyroidism, Ischaemic Heart Disease, Pregnancy, Tuberculosis”

There are conditions that are considered to be risk factors for COVID-19 hospitalisation and death which are not included in this list, i.e. Asthma, chronic liver disease, neurological diseases (stroke/dementia/etc), organ transplant, immunosuppressive condition (rheumatoid arthritis/lupus/etc), in addition to others. For example, see this paper: https://www.nature.com/articles/s41586-020-2521-4#Sec2

3. There are also risk factors such as smoking and obesity that have not been included presumably due to data limitations. It is mentioned in the discussion that obesity could not be included, but I think further discussion on impact of not including them and how this may have affected the results is warranted. (i.e. discussion of the smoking rates/obesity rates)

4. Table 1 - 4 seem to be missing from the manuscript. I have done my best to assess the results without them, but it would be really helpful to see these tables.

5. It is also not completely clear why vaccination could not be included within this analysis. It would be useful to have more of a discussion on the effect of vaccination on hospitalisations and length of stay, and more information on vaccine roll out in South Africa (i.e. % of people vaccinated with 1 or 2 doses during the different waves), and how this would have effected the results observed.

6. I don’t think this is included in the Tables, but I would be interested to see a summary of the different comorbidities (i.e. XX reported Diabetes, XX Cancer; rather than just 0, 1 comorbidities). Perhaps this could be a summary table of the study characteristics. I appreciate this might be a lot of different risk factors and you may have small numbers for some, so perhaps the main comorbidities rather than 0,1,2,>3. Apologies if this is already included within the table 1.

7. Adjusted odds ratios are reported, but it is not clear from the methods exactly what you have adjusted by. This is also true for the reported adusted IRR. I think this could be clarified in the methods (i.e. in the results you say that you did not adjust by medical insurance level for mortality risk as this was highly correlated with age and comorbidities, was this also the case for the hospitalization risk and length of stay/utilization?)

8. I think one of the key strengths of this study is the longer time period over which data was collected, yet this is not really touched on in the discussion. It is not clear to me why mortality and hospitalization would be lower in wave 1? I would have thought that this was before vaccination, and so mortality might have been higher? I think some further discussion would be useful to help readers contextualise these results.

Some minor points:

1. This sentence about the study size in the methods is a little confusing:

“The study population consisted of families (1.7 million) and individuals (3.5 million) belonging to 19 health insurances administered by DH, representing around a third of South Africa’s privately insured

population.”

It is confusing as to whether it is individuals + families, or total individuals, and of these, there are 2.7 families. Maybe it could be re-phrased to make it slightly clearer (if this is the correct meaning):

“The study population consisted of 3.5 million, which included 1.7 million families belonging...”

Also, what is meant by families? Household members sharing the same health insurance policy?

2. In supplementary table 1 – there is a ** after “Rest of the insured population” but no ** underneath the table.

Reviewer #2: COVID-19 hospitalization and mortality and hospitalization-related utilization and expenditure: Analysis of a South African private health insured population.

This paper looking at data from 188000 members of a health insurance in South Africa. The cross sectional study looked at the rates of hospitalisation and death of these insured people.

The value of the study would be to see if the risks, hospitalisation and mortality differed between the public and private sectors in SA.

In the background only relying on systematic reviews is limiting when it comes to studying Africa. Other papers exist and reports etc . for example:

Jassat, W., Cohen, C., Tempia, S., Masha, M., Goldstein, S., Kufa, T., Murangandi, P., Savulescu, D., Walaza, S., Bam, J. L., Davies, M. A., Prozesky, H. W., Naude, J., Mnguni, A. T., Lawrence, C. A., Mathema, H. T., Zamparini, J., Black, J., Mehta, R., Parker, A., … DATCOV author group (2021). Risk factors for COVID-19-related in-hospital mortality in a high HIV and tuberculosis prevalence setting in South Africa: a cohort study. The lancet. HIV, 8(9), e554–e567. https://doi.org/10.1016/S2352-3018(21)00151-X

Mendelsohn AS, De Sá A, Morden E, Botha B, Boulle A, Paleker M, Davies MA. COVID-19 wave 4 in Western Cape Province, South Africa: Fewer hospitalisations, but new challenges for a depleted workforce. S Afr Med J. 2022 Feb 1;112(2):13496. PMID: 35139985.

Van der Westhuizen JN, Hussey N, Zietsman M, et al. Low mortality of people living with diabetes mellitus diagnosed with COVID-19 and managed at a field hospital in Western Cape Province, South Africa. South African medical journal = Suid-Afrikaanse tydskrif vir geneeskunde 2021; 111(10): 961-7.

Phaswana-Mafuya N, Shisana O, Jassat W, et al. Understanding the differential impacts of COVID-19 among hospitalised patients in South Africa for equitable response. South African medical journal = Suid-Afrikaanse tydskrif vir geneeskunde 2021; 111(11): 1084-91.

The other issue missing in the background is that there was (during waves) hospital overcrowding (with private hospitals and /or public hospitals turning people away. – which would impact on mortality)

Costs are also relative and perhaps the Health Market Inquiry should be mentioned which describes over servicing in the private sector.

A spelling mistake on page 12 – Comorbidities

Methods: I am not able to comment on the statistical analysis and suggest that a statistician is consulted.

Results

It would be useful to compare the study population’s demographics to the general population.

Discussion

The hospitalisation and mortality rates vary according to the phase of the epidemic and the improvement in treatment.

It isn’t clear if diabetes with hypertension and heart disease has higher morbidity and mortality or whether it is diabetes with either one?

In paragraph 1 on page 16 – the HIV rates differ, is the number of H positive people too small? 4.7% seems still quite a number to me.– is there an assumption that those with HIV are more compliant? Is there evidence for this.

Page 16 paragraph 2 the other differences between public and private sector relate to testing and diagnosis.

Provincial differences needs more discussion – there are similar hospital groups in different provinces – are their systems different. Were there differences in private hospital access to oxygen, ICU specialists/ expertise, and overcrowding in different provinces?

The average length of stay appears to be longer in the national surveillance system, any ideas on this? It is unclear what exactly is included in the costing.

You should compare the costs to those found in Edoka et al.

Edoka I, Fraser H, Jamieson L, Meyer-Rath G, Mdewa W. Inpatient Care Costs of COVID-19 in South Africa’s Public Healthcare System. Int J Health Policy Manag 2021.)

Which policies will be affected by these results – are they substantially different to those in the public sector and the literature?

6. PLOS authors have the option to publish the peer review history of their article (what does this mean?). If published, this will include your full peer review and any attached files.

Reviewer #1: No

Reviewer #2: No

---

## [Author Response · Author response to Decision Letter 0]

10 Mar 2022

Response to reviewer comments: PONE-D-21-33875

Reviewer 1

I think this is an important study which adds a lot of value, particularly as it covers a region of the world for which there has been limited information. The introduction is well written and interesting. It clearly identifies the research question and highlights the importance of this study. It is a rich dataset, which covers a long period of time, not just the beginning of the pandemic, allowing for changes in hospitalisation, mortality, hospital length and cost to be observed. However, I do have several concerns about the methodology which I have outlined below. Unfortunately, Tables 1-4 also seem to be missing from the manuscript, without which it makes it very difficult to assess the results. I have done my best without them, but would appreciate if a copy of the tables could be sent to me so that I can review the results properly. I also think there lacks interpretation and contextualisation of the results in the discussion (highlighted below).

Response: We apologise that you did not have access to the tables. Due to their size they were uploaded as a separate file which had to be accessed via a link in the article file. We have however now included them in the main article file.

I think if the authors could address these methodological concerns, and expand the interpretation in the discussion, it is a very worthwhile paper.

1. In your population, do you know all people who have received a positive test? Or is it possible that individuals could have received a positive test from a different source, and this is not recorded in the data? This is an important point and should be made clear in the paper. If it is possible that positive tests could have been missed, then this warrants further discussion (i.e. if those individuals did not need hospitalization, this would over-estimate the estimates of hospitalisation and mortality).

Response: As this data set is from a period prior to the mass distribution of home-based COVID-19 self-test kits, COVID testing was by doctor referral available under private insurance at no charge where the patient tested positive. For those paying out of pocket the test was relatively expensive (US$55 per test). It is possible that some patients with private voluntary insurance elected to seek testing in the public system or pay out of pocket for the test in which case they would be missed from the data set. However, we consider that this number would be very small as there was strong incentive for those insured to utilize their insurance benefits, and the requirement for a doctor referral also strengthened capture of data. We further note that (1) our study population was limited to those who tested COVID-19 positive and (2) all the statistics that we present are based on those who tested positive (not the entire insured population). Brief text has been added to the manuscript to reflect this point on page 11.

2. I think there are comorbidities that are known risk factors for COVID-19 which have not been included in the study.

You state in the methods:

“Conditions were considered as comorbidity risk factors for COVID-19 based on a review of published literature: Cancer, Chronic Renal Disease, Congestive Cardiac Failure, Chronic Obstructive Pulmonary Disease, Diabetes Mellitus, HIV, Hypercholesterolaemia, Hypertension, Hypothyroidism, Ischaemic Heart Disease, Pregnancy, Tuberculosis”

There are conditions that are considered to be risk factors for COVID-19 hospitalisation and death which are not included in this list, i.e. Asthma, chronic liver disease, neurological diseases (stroke/dementia/etc), organ transplant, immunosuppressive condition (rheumatoid arthritis/lupus/etc), in addition to others. For example, see this paper: https://www.nature.com/articles/s41586-020-2521-4#Sec2

Response: Thank you for the Williamson et al 2020 reference describing risk factors for COVID-related death that examined 17.3 million patient records in the UK NHS system. We undertook a thorough literature review and consulted with the South African-based medical experts overseeing utilisation management at the health insurance to decide on the final list of co-morbidity risk factors to include as well as consideration of the health profile of private sector patients in South Africa. We therefore do not consider that any major risk factors relevant to our population have been excluded. We have added text to clarify this process on page 6. 

3. There are also risk factors such as smoking and obesity that have not been included presumably due to data limitations. It is mentioned in the discussion that obesity could not be included, but I think further discussion on impact of not including them and how this may have affected the results is warranted. (i.e. discussion of the smoking rates/obesity rates)

Response: Thank you for the comment and we agree that this is a limitation of the available data. As noted we do record this in the limitations section, and have added additional text in relation to smoking status (page 11)

4. Table 1 - 4 seem to be missing from the manuscript. I have done my best to assess the results without them, but it would be really helpful to see these tables.

Response: We apologise that you did not have access to the tables. Due to their size they were uploaded as a separate file which had to be accessed via a link in the article file. We have however now included them in the main article file.

5. It is also not completely clear why vaccination could not be included within this analysis. It would be useful to have more of a discussion on the effect of vaccination on hospitalisations and length of stay, and more information on vaccine roll out in South Africa (i.e. % of people vaccinated with 1 or 2 doses during the different waves), and how this would have affected the results observed.

Response: We have expanded the paragraph in the discussion on vaccination (page 11), and note that the data set timeframe did not correspond with the South African vaccine roll-out. While it would be useful to identify the impact of the vaccine on risk factors, we consider that reporting risks from a “pre-vaccination” timeframe as in our analysis will allow useful comparisons for future analysis. 

6. I don’t think this is included in the Tables, but I would be interested to see a summary of the different comorbidities (i.e. XX reported Diabetes, XX Cancer; rather than just 0, 1 comorbidities). Perhaps this could be a summary table of the study characteristics. I appreciate this might be a lot of different risk factors and you may have small numbers for some, so perhaps the main comorbidities rather than 0,1,2,>3. Apologies if this is already included within the table 1.

Response: Apologies that you were not able to see the tables. Table 4 includes the major co-morbidities and common combinations of co-morbidities and their rates of hospitalisation and death.

7. Adjusted odds ratios are reported, but it is not clear from the methods exactly what you have adjusted by. This is also true for the reported adjusted IRR. I think this could be clarified in the methods (i.e. in the results you say that you did not adjust by medical insurance level for mortality risk as this was highly correlated with age and comorbidities, was this also the case for the hospitalization risk and length of stay/utilization?)

Response: For the multivariable model analyses, we included several well-known measured predictors (e.g. age, gender, province, hospital network, wave) as well health region random effect. Health region was chosen for the random effects to account for potential differences in the population served and the quality of care within different geographic health regions. In the analyses, we performed both univariate and multivariable analyses (including all the predictors and random effects) to identify independent predictors of the modelled outcomes. The multivariate analyses produced adjusted effects as opposed to unadjusted effects from using univariate analysis. This has been added to the analysis methods (page 7) and we have also amended the description of the methods in the abstract and modified reporting of results to help clarify.

8. I think one of the key strengths of this study is the longer time period over which data was collected, yet this is not really touched on in the discussion. It is not clear to me why mortality and hospitalization would be lower in wave 1? I would have thought that this was before vaccination, and so mortality might have been higher? I think some further discussion would be useful to help readers contextualise these results.

Response: Vaccination was not available in South Africa during the period included in this analysis. Hospitalisation and mortality risk were both higher in wave 2 compared to wave 1. This likely reflects the higher case load, health system pressures and the Beta variant in South Africa during wave 2 compared with wave 1. We have included these points in the discussion, page 10.

Some minor points:

1. This sentence about the study size in the methods is a little confusing:

“The study population consisted of families (1.7 million) and individuals (3.5 million) belonging to 19 health insurances administered by DH, representing around a third of South Africa’s privately insured

population.”

It is confusing as to whether it is individuals + families, or total individuals, and of these, there are 2.7 families. Maybe it could be re-phrased to make it slightly clearer (if this is the correct meaning):

“The study population consisted of 3.5 million, which included 1.7 million families belonging...”

Also, what is meant by families? Household members sharing the same health insurance policy?

Response: We have re written this sentence to make the study population clearer (page 5). 

2. In supplementary table 1 – there is a ** after “Rest of the insured population” but no ** underneath the table.

Response: We have revised the column headings to make the table clearer.

Reviewer #2: 

This paper looking at data from 188000 members of a health insurance in South Africa. The cross sectional study looked at the rates of hospitalisation and death of these insured people.

The value of the study would be to see if the risks, hospitalisation and mortality differed between the public and private sectors in SA.

In the background only relying on systematic reviews is limiting when it comes to studying Africa. Other papers exist and reports etc . for example:

Jassat, W., Cohen, C., Tempia, S., Masha, M., Goldstein, S., Kufa, T., Murangandi, P., Savulescu, D., Walaza, S., Bam, J. L., Davies, M. A., Prozesky, H. W., Naude, J., Mnguni, A. T., Lawrence, C. A., Mathema, H. T., Zamparini, J., Black, J., Mehta, R., Parker, A., … DATCOV author group (2021). Risk factors for COVID-19-related in-hospital mortality in a high HIV and tuberculosis prevalence setting in South Africa: a cohort study. The lancet. HIV, 8(9), e554–e567. https://doi.org/10.1016/S2352-3018(21)00151-X

Mendelsohn AS, De Sá A, Morden E, Botha B, Boulle A, Paleker M, Davies MA. COVID-19 wave 4 in Western Cape Province, South Africa: Fewer hospitalisations, but new challenges for a depleted workforce. S Afr Med J. 2022 Feb 1;112(2):13496. PMID: 35139985.

Van der Westhuizen JN, Hussey N, Zietsman M, et al. Low mortality of people living with diabetes mellitus diagnosed with COVID-19 and managed at a field hospital in Western Cape Province, South Africa. South African medical journal = Suid-Afrikaanse tydskrif vir geneeskunde 2021; 111(10): 961-7.

Phaswana-Mafuya N, Shisana O, Jassat W, et al. Understanding the differential impacts of COVID-19 among hospitalised patients in South Africa for equitable response. South African medical journal = Suid-Afrikaanse tydskrif vir geneeskunde 2021; 111(11): 1084-91.

Response: Thank you for alerting us to these articles. Three of these articles were published after we submitted our paper to Plos One (on 27 October 2021) and we had referenced another article from Jassat presenting data from the same cohort study (reference 14). We have now included reference to the suggested papers where we feel they have relevance. 

The other issue missing in the background is that there was (during waves) hospital overcrowding (with private hospitals and /or public hospitals turning people away. – which would impact on mortality)

Response: Thank you for your comment. We agree that overcrowding has been a particularly relevant issue for the management of COVID-19, however this is a complex dynamic with uncertainty regarding the actual impact on death and COVID-19 outcomes and the distinction between delayed or deferred admission and/or admission to alternative facilities. Critically for this analysis, it is uncertain whether overcrowding had a differential impact on admission and death contingent upon particular risk factors. We consider that inclusion of the different pandemic waves in our models would incorporate the impact of overcrowding. We consider that this point is more suited for the discussion rather than the background and have add text in the discussion to reflect this point (page 11).

Costs are also relative and perhaps the Health Market Inquiry should be mentioned which describes over servicing in the private sector.

Response: We have made reference to the health market enquiry in the discussion in relation to private sector costs. We feel this is best placed in the discussion section (page 12).

A spelling mistake on page 12 – Comorbidities

Response: We have corrected this word throughout with no hyphen.

Methods: I am not able to comment on the statistical analysis and suggest that a statistician is consulted.

Response: Professor Manda is a statistician and has undertaken all of the statistical analysis.

Results

It would be useful to compare the study population’s demographics to the general population.

Response: We have expanded on the description in the background of the differences between the general population reliant on the public health sector and the private insured population in South Africa (page 4)

Discussion

The hospitalisation and mortality rates vary according to the phase of the epidemic and the improvement in treatment.

Response: we have included text in the discussion regarding differences in outcomes between pandemic waves (page 10).

It isn’t clear if diabetes with hypertension and heart disease has higher morbidity and mortality or whether it is diabetes with either one?

Response: thank you we have re written that sentence. It is the combination of diabetes, hypertension, hypercholesterolemia and ischaemic heart disease that carried the greatest risk of hospitalisation and mortality (page 10).

In paragraph 1 on page 16 – the HIV rates differ, is the number of H positive people too small? 4.7% seems still quite a number to me.– is there an assumption that those with HIV are more compliant? Is there evidence for this.

Response: The difference in HIV prevalence between patients attending the private and public sector in South Africa is well described and is related to underlying social and economic determinants of vulnerability to HIV. We have cited another study from South Africa that shows a similar difference between HIV prevalence amongst public and private sector patients admitted with COVID-19 (page 10)

Page 16 paragraph 2 the other differences between public and private sector relate to testing and diagnosis.

Response: We do not expect rates of testing in the public and private sector to have an influence on mortality amongst hospitalised patients. Certainly rates of testing would influence COVID-19 case numbers but should not influence mortality. 

Provincial differences needs more discussion – there are similar hospital groups in different provinces – are their systems different. Were there differences in private hospital access to oxygen, ICU specialists/ expertise, and overcrowding in different provinces?

Response: we have added further discussion on possible reasons for the provincial differences in outcomes (page 10).

The average length of stay appears to be longer in the national surveillance system, any ideas on this? It is unclear what exactly is included in the costing.

Response: We have not found any published data on length of hospital stay in the national surveillance system that corresponds to the period that we considered. We have expanded on the description of costing data on page 6.

You should compare the costs to those found in Edoka et al.

Edoka I, Fraser H, Jamieson L, Meyer-Rath G, Mdewa W. Inpatient Care Costs of COVID-19 in South Africa’s Public Healthcare System. Int J Health Policy Manag 2021.)

Response: Thank you for the Edoka et al paper which we are aware of and which estimated hospitalization costs in the public sector using a bottom up/micro approach. The Edoka paper is a methodologically robust analysis and an important contribution to the literature but there is very little comparability between private sector provider fees charged to insurers and a bottom up estimation of “cost of provision” in the public sector. The reasons for this lack of comparability are important and are a unique area of research in costing methodology but are not the subject of this paper. Of particular importance within this paper is the variation of fees charged between providers for seemingly similar cases in the same context. 

Which policies will be affected by these results – are they substantially different to those in the public sector and the literature?

Response: Thank you for the query. We have noted the various policy implications for the results including stratification by risk, reducing unwarranted variation and larger regulatory reform in terms of cost containment and improving efficiency. Some policy implications are implicitly targeted towards private sector (such as need for regulatory reform) but more general concepts relating to differentiated care based on risk are applicable to both public and private sectors. We are open to making revisions to text in specific instances to make this clearer and have reviewed the text but cannot identify any areas where this can be made more explicit. 

---

## [Decision Letter · Decision Letter 1]

21 Apr 2022

COVID-19 hospitalization and mortality and hospitalization-related utilization and expenditure: Analysis of a South African private health insured population.

PONE-D-21-33875R1

Dear Dr. Doherty,

We’re pleased to inform you that your manuscript has been judged scientifically suitable for publication and will be formally accepted for publication once it meets all outstanding technical requirements.

Kind regards,

Raymond Nienchen Kuo, Ph.D

Academic Editor

PLOS ONE

Reviewers' comments:

Reviewer's Responses to Questions

**Comments to the Author**

1. If the authors have adequately addressed your comments raised in a previous round of review and you feel that this manuscript is now acceptable for publication, you may indicate that here to bypass the “Comments to the Author” section, enter your conflict of interest statement in the “Confidential to Editor” section, and submit your "Accept" recommendation.

Reviewer #1: All comments have been addressed

Reviewer #2: All comments have been addressed

2. Is the manuscript technically sound, and do the data support the conclusions?

Reviewer #1: Yes

Reviewer #2: Yes

3. Has the statistical analysis been performed appropriately and rigorously? 

Reviewer #1: Yes

Reviewer #2: Yes

4. Have the authors made all data underlying the findings in their manuscript fully available?

Reviewer #1: No

Reviewer #2: Yes

5. Is the manuscript presented in an intelligible fashion and written in standard English?

Reviewer #1: Yes

Reviewer #2: Yes

6. Review Comments to the Author

Reviewer #1: (No Response)

Reviewer #2: I acknowledge and am happy with the changes the authors have made. The paper is ready for publication.

7. PLOS authors have the option to publish the peer review history of their article (what does this mean?). If published, this will include your full peer review and any attached files.

Reviewer #1: No

Reviewer #2: No

---

## [Editor Report · Acceptance letter]

27 Apr 2022

PONE-D-21-33875R1 

COVID-19 hospitalization and mortality and hospitalization-related utilization and expenditure:  Analysis of a South African private health insured population. 

Dear Dr. Doherty:

I'm pleased to inform you that your manuscript has been deemed suitable for publication in PLOS ONE. Congratulations! Your manuscript is now with our production department. 

Kind regards, 

on behalf of

Professor Raymond Nienchen Kuo 

Academic Editor

PLOS ONE